# Joint Policy Search for Multi-agent Collaboration with Imperfect Information

**Yuandong Tian**
Facebook AI Research
yuandong@fb.com

**Qucheng Gong**
Facebook AI Research
qucheng@fb.com

**Tina Jiang**
Facebook AI Research
tinayujiang@fb.com

## Abstract

To learn good joint policies for multi-agent collaboration with imperfect information remains a fundamental challenge. While for two-player zero-sum games, coordinate-ascent approaches (optimizing one agent's policy at a time, e.g., self-play [35, 20]) work with guarantees, in multi-agent cooperative setting they often converge to sub-optimal Nash equilibrium. On the other hand, directly modeling joint policy changes in imperfect information game is nontrivial due to complicated interplay of policies (e.g., upstream updates affect downstream state reachability). In this paper, we show global changes of game values can be decomposed to policy changes localized at each information set, with a novel term named *policy-change density*. Based on this, we propose *Joint Policy Search* (JPS) that iteratively improves joint policies of collaborative agents in imperfect information games, without re-evaluating the entire game. On multi-agent collaborative tabular games, JPS is proven to never worsen performance and can improve solutions provided by unilateral approaches (e.g, CFR [44]), outperforming algorithms designed for collaborative policy learning (e.g. BAD [16]). Furthermore, for real-world game with exponential states, JPS has an online form that naturally links with gradient updates. We test it to Contract Bridge, a 4-player imperfect-information game where a team of 2 collaborates to compete against the other. In its bidding phase, players bid in turn to find a good contract through a limited information channel. Based on a strong baseline agent that bids competitive Bridge purely through domain-agnostic self-play, JPS improves collaboration of team players and outperforms WBridge5, a championship-winning software, by +0.63 IMPs (International Matching Points) per board over 1000 games, substantially better than previous SoTA (+0.41 IMPs/b against WBridge5) under Double-Dummy evaluation. Note that +0.1 IMPs/b is regarded as a nontrivial improvement in Computer Bridge. Part of the code is released in https://github.com/facebookresearch/jps.

## 1 Introduction

Deep reinforcement learning has demonstrated strong or even super-human performance in many complex games (e.g., Atari [28], Dota 2 [30], Starcraft [42], Poker [5, 29], Find and Seek [1], Chess, Go and Shogi [34, 36, 39]). While massive computational resources are used, the underlying approach is quite simple: to iteratively improve the policy of the current agent, assuming stationary environment and fixed policies of all other agents. Although for two-player zero-sum games this is effective, for multi-agent collaborative with imperfect information, it often leads to sub-optimal Nash equilibria where none of the agents is willing to change their policies unilaterally. For example, if speaking one specific language becomes a convention, then unilaterally switching to a different one is not a good choice, even if the other agent actually knows that language better.

In this case, it is necessary to learn to jointly change policies of multiple agents to achieve better equilibria. One brute-force approach is to change policies of multiple agents simultaneously, and re-evaluate them one by one on the entire game to seek for performance improvement, which is

computationally expensive. Alternatively, one might hope that a change of a sparse subset of policies might lead to "local" changes of game values and evaluating these local changes can be faster. While this is intuitively reasonable, in imperfect information game (IG), a local policy change could affects the value of both downstream and upstream decision points, leading to non-local interplay.

In this paper, we realize this locality idea by proposing *policy-change density*, a quantity defined at each perfect information history state with two key properties: **(1)** when summing over all states, it gives overall game value changes upon policy update, and **(2)** when the local policy remains the same, the density vanishes regardless of any policy changes at other parts of the game tree. Based on this density, the value changes of any policy update on a sparse set of decision points can be decomposed into a summation on each decision point (or information set), which is easy and efficient to compute.

Based on that, we propose a novel approach, called *Joint Policy Search* (JPS). For tabular IG, JPS is proven to never worsen the current policy, and is computationally more efficient than brute-force approaches. For simple collaborative games with enumerable states, we show that JPS improves policies returned by Counterfactual Regret Minimization baseline [44] by a fairly good margin, outperforming methods with explicit belief-modeling [16] and Advantageous Actor-Critic (A2C) [27] with self-play, in particular in more complicated games.

Furthermore, we show JPS has a sample-based formulation and can be readily combined with gradient methods and neural networks. This enables us to apply JPS to Contract Bridge bidding, in which enumerating the information sets are computationally prohibitive[1]. Improved by JPS upon a strong A2C baseline, the resulting agent outperforms Wbridge5, a world computer bridge program that won multiple championships, by a large margin of $+0.63$ IMPs per board (*IMPs/b*) over a tournament of 1000 games, better than previous state-of-the-art [18] that beats WBridge5 by $+0.41$ *IMPs/b*. All of them use Double-Dummy evaluation [19]. Note that $+0.1$ *IMPs/b* is regarded as nontrivial improvement in computer bridge [32].

## 2 Related work

**Methods to Solve Extensive-Form Games**. For two-player zero-sum extensive-form games, many algorithms have been proposed with theoretical guarantees. For perfect information game (PG), $\alpha$-$\beta$ pruning, Iterative Deepening depth-first Search [21], Monte Carlo Tree Search [13] are used in Chess [9] and Go [34, 40], yielding strong performances. For imperfect information games (IG), Double-Oracle [26], Fictitious (self-)play [20] and Counterfactual Regret Minimization (CFR [44, 23]) can be proven to achieve Nash equilibrium. These algorithms are *coordinate-ascent*: iteratively find a best response to improve the current policy, given the opponent policies over the history.

On the other hand, it is NP-hard to obtain optimal policies for extensive-form collaborative IG where two agents collaborate to achieve a best common pay-off [10]. Such games typically have multiple sub-optimal Nash equilibria, where unilateral policy update cannot help [14]. Many empirical approaches have been used. Self-play was used in large-scale IG that requires collaboration like Dota 2 [30] and Find and Seek [1]. Impressive empirical performance is achieved with huge computational efforts. Previous works also model belief space (e.g., Point-Based Value Iteration [31] in POMDP, BAD [16]) or model the behaviors of other agents (e.g., AWESOME [11], Hyper Q-learning [37], LOLA [15]). To our best knowledge, we are the first to propose a framework for efficient computation of policy improvement of multi-agent collaborative IG, and show that it can be extended to a sample-based form that is compatible with gradient-based methods and neural networks.

**Solving Imperfect Information Games**. While substantial progress has been made in PG, how to effectively solve IG in general remains open. Libratus [5] and Pluribus [6] outperform human experts in two-player and multi-player no-limit Texas Holdem with CFR and domain-specific state abstraction, and DeepStack [29] shows expert-level performance with continual re-solving. ReBeL [3] adapts AlphaZero style self-play to IIG, achieving superhuman level in Poker with much less domain knowledge. Recently, [24] shows strong performance in Hanabi using collaborative search with a pre-defined common blueprint policy. Suphx [25] achieves superhuman level in Mahjong with supervised learning and policy gradient. DeepRole achieves superhuman level [33] in *The Resistance: Avalon* with continual re-solving [29].

In comparison, Contract Bridge with team collaboration, competition and a huge space of hidden information, remains unsolved. While the playing phase has less uncertainty and champions of

computer bridge tournament have demonstrated strong performances against top professionals (e.g., GIB [17], Jack [22], Wbridge5 [12]), bidding phase is still challenging due to much less public information. Existing software hard-codes human bidding rules. Recent works [43, 32, 18] use DRL to train a bidding agent, which we compare with. See Sec. 5 for details.

## 3 Background and Notation

In this section, we formulate our framework in the more general setting of general-sum games, where each of the $C$ players could have a different reward. In this paper, our technique is mainly applied to pure collaborative IGs and we leave its applications in other types of games for future work.

Let $h$ be a perfect information state (or **state**) of the game. From game start, $h$ is reached via a sequence of public and private actions: $h = a_1 a_2 \ldots a_d$ (abbreviated as $a_{\leq d}$). $I = \{h\}$ is an information set (or **infoset**) that contains all perfect states indistinguishable from the current player's point of view (e.g., in Poker, $I$ hold all possibilities of opponent cards given public cards and the player's private cards). All $h \in I$ share the same policy $\sigma(h) = \sigma(I)$ and $\sigma(I, a)$ is the probability of taking action $a$. $A(I)$ is the set of allowable actions for infoset $I$.

Let $I(h)$ be the infoset associated with state $h$. $ha$ is the unique next state after taking action $a$ from $h$. $h'$ is a **descendant** of $h$, denoted as $h \sqsubset h'$, if there exists a sequence of actions $\{a_1, a_2, \ldots, a_d\}$ so that $h' = ha_1 a_2 \ldots a_d = ha_{\leq d}$. The **successor set** $\mathrm{succ}(I, a)$ contains all the next infosets after taking action $a$ from $I$. The size of $\mathrm{succ}(I, a)$ can be large (e.g., the opponent/partner can make many different decisions based on her private cards). The **active set** $\mathcal{I}(\sigma', \sigma) := \{I : \sigma'(I) \neq \sigma(I)\}$ is the collection of infosets where the policy differs between $\sigma$ and $\sigma'$.

$\pi^\sigma(h) := \prod_{i=1}^{d-1} \sigma(I(a_{<i}), a_i)$ is the **reachability**: the probability of reaching state $h = a_1 a_2 \ldots a_d$ following the policy $\sigma$. Note that unlike CFR [44], we use *total* reachability: it includes the probability incurred by chance (or nature) actions and other player's actions under current policy $\sigma$. $Z$ is the **terminal set**. Each terminal state $z \in Z$ has a reward (or utility) $\boldsymbol{r}(z) \in \mathbb{R}^C$, where $C$ is the number of players. The $i$-th element of $\boldsymbol{r}(z)$, $r_i(z)$, is the pay-off of the $i$-th player.

For state $h \notin Z$, its **value function** $\boldsymbol{v}^\sigma(h) \in \mathbb{R}^C$ under the current policy $\sigma$ is:

$$\boldsymbol{v}^\sigma(h) = \sum_{a \in A(I(h))} \sigma(I(h), a) \boldsymbol{v}^\sigma(ha) \tag{1}$$

For terminal node $h \in Z$, its value $\boldsymbol{v}^\sigma(z) = \boldsymbol{v}(z) = \boldsymbol{r}(z)$ is independent of the policy $\sigma$. Intuitively, the value function is the expected reward starting from state $h$ following $\sigma$.

For IG, what we can observe is infoset $I$ but not state $h$. Therefore we could define **macroscopic** reachability $\pi^\sigma(I) = \sum_{h \in I} \pi^\sigma(h)$, value function $\boldsymbol{v}^\sigma(I)$ and $Q$-function $\boldsymbol{q}^\sigma(I, a)$:

$$\boldsymbol{v}^\sigma(I) = \sum_{h \in I} \pi^\sigma(h) \boldsymbol{v}^\sigma(h), \qquad \boldsymbol{q}^\sigma(I, a) = \sum_{h \in I} \pi^\sigma(h) \boldsymbol{v}^\sigma(ha) \tag{2}$$

and their conditional version: $\boldsymbol{V}^\sigma(I) = \boldsymbol{v}^\sigma(I)/\pi^\sigma(I)$ and $\boldsymbol{Q}^\sigma(I, a) = \boldsymbol{q}^\sigma(I, a)/\pi^\sigma(I)$. If we train DRL methods like DQN [28] and A3C [27] on IG without a discount factor, $\boldsymbol{V}^\sigma(I)$ and $\boldsymbol{Q}^\sigma(I, a)$ are the terms actually learned in neural networks. As one key difference between PG and IG, $\boldsymbol{v}^\sigma(h)$ only depends on the *future* of $\sigma$ after $h$ but $\boldsymbol{V}^\sigma(I)$ also depends on the *past* of $\sigma$ before $h$ due to involved reachability. This is because other players' policies affect the reachability of states $h$ within the current infoset $I$, which is invisible to the current player.

Finally, we define $\bar{\boldsymbol{v}}^\sigma \in \mathbb{R}^C$ as the overall game value for all $C$ players. $\bar{\boldsymbol{v}}^\sigma := \boldsymbol{v}^\sigma(h_0)$ where $h_0$ is the game start (before any chance node, e.g., card dealing).

## 4 A Theoretical Framework for Evaluating Local Policy Change

We start with a novel formulation to evaluate *local* policy change, which means that the active set $\mathcal{I}(\sigma, \sigma') = \{I : \sigma(I) \neq \sigma'(I)\}$ is much smaller than the total number of infosets. A naive approach is to evaluate the new policy $\sigma'$ over the entire game tree, which is computationally expensive.

One might wonder for each policy proposal $\sigma'$, is that possible to decompose $\bar{\boldsymbol{v}}^{\sigma'} - \bar{\boldsymbol{v}}^\sigma$ onto each individual infoset $I \in \mathcal{I}(\sigma, \sigma')$. However, unlike PG, due to interplay of upstream policies with downstream reachability, a local change of policy affects the utility of its downstream states. For example, a trajectory might leave an active infoset $I_1$ and and later re-enter another active infoset

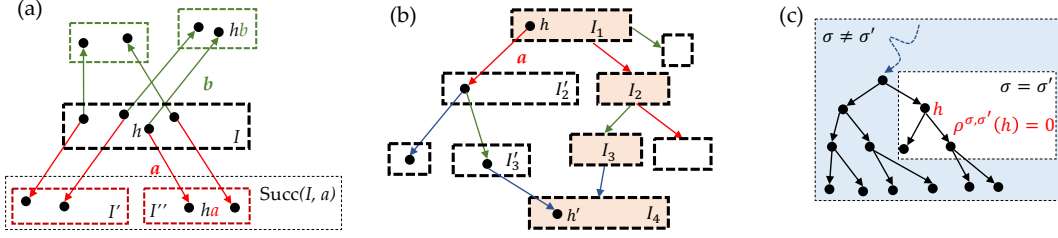

Figure 1: **(a)** Basic notations. **(b)** Hard case: a perfect information state $h'$ could first leave active infoset $I_1$, then re-enter the infoset (at $I_4$). Note that it could happen in perfect-recall games, given all the public actions are the same (shown in common red, green and blue edges) and $I_2$ and $I_4$ are played by different players. **(c)** Our formulation defines *policy-change density* $\boldsymbol{\rho}^{\sigma,\sigma'}$ that vanishes in regions with $\sigma' = \sigma$, regardless of its upstream/downstream context where $\sigma' \neq \sigma$.

$I_4$ (Fig. 1(b)). In this case, the policy change at $I_1$ affects the evaluation on $I_4$. Such long-range interactions can be quite complicated to capture.

This decomposition issue in IG have been addressed in many previous works (e.g., CFR-D [8, 7], DeepStack [29], Reach subgame solving [4]), mainly in the context of solving subgames in a principled way in two-player zero-sum games (like Poker). In contrast, our framework allows simultaneous policy changes at different parts of the game tree, even if they could be far apart, and can work in general-sum games. To our best knowledge, no framework has achieved that so far.

In this section, we coin a novel quantity called *policy-change density* to achieve this goal.

## 4.1 A Localized Formulation

We propose a novel formulation to *localize* such interactions. For each state $h$, we first define the following *cost* $\boldsymbol{c}^{\sigma,\sigma'} \in \mathbb{R}^C$ and *policy-change density* $\boldsymbol{\rho}^{\sigma,\sigma'} \in \mathbb{R}^C$:

$$\boldsymbol{c}^{\sigma,\sigma'}(h) = (\pi^{\sigma'}(h) - \pi^{\sigma}(h))\boldsymbol{v}^{\sigma}(h), \qquad \boldsymbol{\rho}^{\sigma,\sigma'}(h) = -\boldsymbol{c}^{\sigma,\sigma'}(h) + \sum_{a \in A(h)} \boldsymbol{c}^{\sigma,\sigma'}(ha) \quad (3)$$

Intuitively, $\boldsymbol{c}^{\sigma,\sigma'}(h)$ means if we switch from $\sigma$ to $\sigma'$, what would be the difference in terms of expected reward, if the new policy $\sigma'$ remains the same for all $h$'s descendants. For policy-change density $\boldsymbol{\rho}^{\sigma,\sigma'}$, the intuition behind its name is clear with the following lemmas:

**Lemma 1** (Density Vanishes if no Local Policy Change). *For $h$, if $\sigma'(h) = \sigma(h)$, then $\boldsymbol{\rho}^{\sigma,\sigma'}(h) = \boldsymbol{0}$.*

**Lemma 2** (Density Summation). *$\bar{\boldsymbol{v}}^{\sigma'} - \bar{\boldsymbol{v}}^{\sigma} = \sum_{h \notin Z} \boldsymbol{\rho}^{\sigma,\sigma'}(h)$.*

Intuitively, Lemma 1 shows that $\boldsymbol{\rho}^{\sigma,\sigma'}$ vanishes if policy does not change *within* a state, regardless of whether policy changes in other part of the game. As a result, $\boldsymbol{\rho}^{\sigma,\sigma'}$ **is a *local* quantity with respect to policy change**. In comparison, quantities like $\pi^{\sigma}$, $v^{\sigma}$, $c$ and $\pi^{\sigma'}v^{\sigma'} - \pi^{\sigma}v^{\sigma}$ are *non-local*: e.g., $\boldsymbol{v}^{\sigma}(h)$ (or $\pi^{\sigma}(h)$) changes if the downstream $\boldsymbol{v}^{\sigma}(h')$ (or upstream $\boldsymbol{v}^{\sigma}(h')$) changes due to $\sigma \to \sigma'$, even if the local policy remains the same (i.e., $\sigma(h) = \sigma'(h)$).

With this property, we now address how to decompose $\bar{\boldsymbol{v}}^{\sigma'} - \bar{\boldsymbol{v}}^{\sigma}$ onto active set $\mathcal{I}$. According to Lemma 1, for any infoset $I$ with $\sigma'(I) = \sigma(I)$, the policy-change density vanishes. From Lemma 2, the summation of density over the entire subtree is exactly the overall value difference due to the policy change. If we put both together, we get:

**Theorem 1** (InfoSet Decomposition of Policy Change). *When $\sigma \to \sigma'$, the change of game value is:*

$$\bar{\boldsymbol{v}}^{\sigma'} - \bar{\boldsymbol{v}}^{\sigma} = \sum_{I \in \mathcal{I}} \sum_{h \in I} \boldsymbol{\rho}^{\sigma,\sigma'}(h) \quad (4)$$

Theorem 1 is the main theorem that decomposes local policy changes to each infoset in the active set $\mathcal{I}$. We will see how it is utilized to find a better policy $\sigma'$ from the existing one $\sigma$.

## 4.2 Comparison with regret in CFR

From Eqn. 3, we could rewrite the density $\boldsymbol{\rho}^{\sigma,\sigma'}(h)$ in a more concise form after some algebraic manipulation:

$$\boldsymbol{\rho}^{\sigma,\sigma'}(h) = \pi^{\sigma'}(h)\left[\sum_{a \in A(I)} \sigma'(I,a)\boldsymbol{v}^{\sigma}(ha) - \boldsymbol{v}^{\sigma}(h)\right] \quad (5)$$

We conceptually compare our $\boldsymbol{\rho}^{\sigma,\sigma'}(h)$ with the regret term in vanilla CFR [44] (Eqn. 7), which takes the form of $\pi^{\sigma}_{-i}(h)(\boldsymbol{v}^{\sigma}(ha) - \boldsymbol{v}^{\sigma}(h))$ for player $i$ who is to play action $a$ at infoset $I(h)$. The key difference here is that our $\boldsymbol{\rho}^{\sigma,\sigma'}(h)$ uses the total reachability $\pi^{\sigma'}(h)$ evaluated on the *new* policy $\sigma'$, while CFR uses the except-player-$i$ reachability $\pi^{\sigma}_{-i}(h)$ evaluated on the *old* policy $\sigma$.

We emphasize that this small change leads to very different (and novel) theoretical insights. It leads to our policy-change decomposition (Theorem 1) that *exactly* captures the value difference before and after policy changes for general-sum games, while in CFR, summation of the regret at each infoset $I$ is *an upper bound* of the Nash exploitability for two-player zero-sum games. Our advantage comes with a price: the regret in CFR only depends on the old policy $\sigma$ and can be computed independently at each infoset, while computing our policy-change density requires a re-computation of the altered reachability due to new policy $\sigma'$ on the upstream infosets, which will be addressed in Sec. 5.1. From the derivation, we could also see that in CFR, the assumption of *perfect recall* is needed to ensure that no double counting exists so that the upper bound can hold (Eqn. 15 in [44]), while in our Eqn. 5, there is no such requirement.

# 5 Joint Policy Search for Pure Collaborative Multi-agent IGs

In this paper, we focus on pure collaborative games, in which all players share the *common* value. Hence, we replace $\boldsymbol{\rho}^{\sigma,\sigma'}$ with scalar $\rho^{\sigma,\sigma'}$ (similar for $v^{\sigma}$). More general cases are left in future work. For pure collaborative settings, we propose *Joint Policy Search* (JPS), a novel approach to jointly optimize policies of multiple agents at the same time in IG. Our goal is to find a policy improvement $\sigma'$ so that it is guaranteed that the changes of *overall game value* $\bar{v}^{\sigma'} - \bar{v}^{\sigma}$ is always non-negative.

---

**Algorithm 1** Joint Policy Search (Tabular form)

---

1: **function** JSP-MAIN($\sigma$)
2:     **for** $i = 1 \ldots T$ **do**
3:         Compute reachability $\pi^{\sigma}$ and value $v^{\sigma}$ under $\sigma$. Pick initial infoset $I_1$.
4:         $\sigma \leftarrow$ JPS($\sigma, \{I_1\}, 1$).
5:     **end for**
6: **end function**
7: **function** JPS($\sigma, \mathcal{I}_{\text{cand}}, d$)                                            $\triangleright$ $\mathcal{I}_{\text{cand}}$: *candidate infosets*
8:     **if** $d \geq D$ **then**
9:         **return** 0.                                           $\triangleright$ *Search reaches maximal depth $D$*
10:     **end if**
11:     **for** $I \in \mathcal{I}_{\text{cand}}$ and $h \in I$ **do**
12:         Compute $\pi^{\sigma'}(h)$ by back-tracing $h' \sqsubset h$ until $I(h')$ is active. Otherwise $\pi^{\sigma'}(h) = \pi^{\sigma}(h)$.
13:     **end for**
14:     Compute $J^{\sigma,\sigma'}(I) = \sum_{h \in I} \rho^{\sigma,\sigma'}(h)$ for each $I \in \mathcal{I}_{\text{cand}}$ using Eqn. 5.
15:     **for** $I \in \mathcal{I}_{\text{cand}}$ and $a \in A(I)$ **do**
16:         Set $I$ active. Set $\sigma'(I)$ and reachability accordingly Eqn. 6.
17:         Set $r(I, a) = $ JPS($\sigma, \text{succ}(I, a), d+1$) $+ J^{\sigma,\sigma'}(I)$
18:     **end for**
19:     **return** $\max(0, \max_{I,a} r(I, a))$                    $\triangleright$ *Also consider if no infoset in $\mathcal{I}_{\text{cand}}$ is active.*
20: **end function**

---

## 5.1 Joint Policy Search (JPS)

Using Theorem 1, we now can evaluate $\bar{v}^{\sigma'} - \bar{v}^{\sigma}$ efficiently given an active set $\mathcal{I}$.

Based on that, a naive way for policy improvement, is to first pick and fix an active set $\mathcal{I}$, and then (jointly) optimize the policies $\sigma(I)$ on each $I \in \mathcal{I}$. In contrast, our JPS uses a different strategy: it first proposes a new policy at the current infoset, and then *dynamically* construct new active infosets to focus on. The underlying motivation is that once a upstream policy on infoset $I$ has changed, the downstream policies on $\text{succ}(I)$ often need to be changed as well, in particular when two agents playing consequence action need to coordinate to jump out of local equilibrium.

This naturally leads to a depth-first search algorithm (Alg. 1). We first pick $\mathcal{I} = \{I_1\}$ where $I_1$ is a "root" infoset, change the policy of $I_1$ (decision ①), then pick an action $a_1 \in A(I_1)$ (decision ②), and pick an infoset $I_2 \in \text{succ}(I_1, a_1)$ into the active set $\mathcal{I}$ (decision ③), and goes down into the game tree until a maximal depth has been reached. When the maximal depth $D$ is reached, we have constructed an active set $\mathcal{I} = [I_1, \ldots, I_D]$ so that $I_{i+1} \in \text{succ}(I_i, a_i)$ with some $a_i$ and Theorem 1 can be applied to compute the policy improvement. We then backtrace over all the decision points (①, ② and ③) and find the best policy change $\sigma'$ that leads to the best improvement $\bar{v}^{\sigma'} - \bar{v}^{\sigma}$. Such a procedure can be repeated for $T$ iterations to obtain final improved policy (JSP-Main in Alg. 1).

For each infoset $I$, any policy change $\sigma'$ could work. Here we use one-hot policy and merge the decision point ① and ② for search efficiency:

$$\sigma'(I_i, a) = \mathbb{I}[a = a_i] \tag{6}$$

Note that for pure collaborative games, we don't consider mixed strategies since they are dominated by pure strategies. To compute $\rho^{\sigma,\sigma'}$, before the search starts, we first sweep all the states $h$ to get $v^{\sigma}(h)$ and $\pi^{\sigma}(h)$, which can be shared across different search branches. During search, the only term we need to recompute for different search branches is the altered reachability $\pi^{\sigma'}$, which depends on upstream policy changes. Note that since we use depth-first search, the upstream policy change is always available and can be easily retrieved. The search has the complexity of $\mathcal{O}(|S| + M)$, where $|S|$ is the total number of states and $M$ is the number of policy candidates. This is more efficient than brute-force search that requires a complete sweep of all states for each policy candidate ($\mathcal{O}(|S|M)$).

**Theorem 2** (Performance Guarantee for Alg. 1). $\bar{v}^{\sigma'} \geq \bar{v}^{\sigma}$ for $\sigma' = $ JSP-Main$(\sigma)$.

## 5.2 Online Joint Policy Search (OJPS)

To compute quantities in Theorem 1, we still need to compute $\pi^{\sigma}$ and $v^{\sigma}$ on all states. This makes it hard for real-world scenarios (e.g., Contract Bridge), where an enumeration of all states is computationally infeasible. Therefore, we consider an online sampling version. Define $J^{\sigma,\sigma'}(I) = \sum_{h \in I} \rho^{\sigma,\sigma'}(h)$ and $J$ can be decomposed into two terms $J(I) = J_1(I) + J_2(I)$ ($\lambda$ is a constant):

$$J_1(I) = \sum_{h \in I} (\pi^{\sigma'}(h) - \lambda\pi^{\sigma}(h)) \left( \sum_{a \in A(I)} \sigma'(I, a)v^{\sigma}(ha) - v^{\sigma}(h) \right), \quad J_2(I) = \lambda \sum_{h \in I} \pi^{\sigma}(h) \left( \sum_{a \in A(I)} \sigma'(I, a)Q^{\sigma}(I, a) - V^{\sigma}(I) \right) \tag{7}$$

If we sample a trajectory by running the current policy $\sigma$ and pick one perfect information state $h_0$, then $h_0 \sim \pi^{\sigma}(\cdot)$. Then, for $I = I(h)$, using this sample $h_0$, we can compute $\hat{J}_1(I) = (\pi^{\sigma'}(h|h_0) - \lambda\pi^{\sigma}(h|h_0))(\sum_a \sigma'(I, a)v^{\sigma}(ha) - v^{\sigma}(h))$ and $\hat{J}_2(I) = \lambda\pi^{\sigma}(h|h_0)(\sum_a \sigma'(I, a)Q^{\sigma}(I, a) - V^{\sigma}(I))$ can be computed via macroscopic quantities (eg., from neural network). Here $\pi^{\sigma}(h|h_0) := \pi^{\sigma}(h)/\pi^{\sigma}(h_0)$ is the (conditional) probability of reaching $h$ starting from $h_0$. Intuitively, $\hat{J}_1$ accounts for the benefits of taking actions that favors the current state $h$ (e.g., what is the best policy if all cards are public?), and $\hat{J}_2$ accounts for effects due to other perfect information states that are not yet sampled. The hyper-parameter $\lambda$ controls their relative importance. Therefore, it is possible that we could use a few perfect information states $h$ to improve imperfect information policy via searching over the best sequence of joint policy change. The resulting action sequence representing joint policy change is sent to the replay buffer for neural network training.

## 6 Experiments on Simple Collaborative Games

We try JPS on multiple simple two-player pure collaborative IGs to demonstrate its effectiveness. Except for private card dealing, all actions in these games are public knowledge with perfect recall. Note that JPS can be regarded as a booster to improve any solutions from any existing approaches.

**Definition 1** (Simple Communication Game of length $L$). *Consider a game where $s_1 \in \{0, \ldots, 2^L - 1\}$, $a_1 \in \mathcal{A}_1 = \{0, 1\}$, $a_2 \in \mathcal{A}_2 \in \{0, \ldots, 2^L - 1\}$. P1 sends one binary public signal for $L$ times, then P2 guess P1's private $s_1$. The reward $r = \mathbf{1}[s_1 = a_2]$ (i.e. 1 if guess right).*

**Definition 2** (Simple Bidding Game of size $N$). *P1 and P2 each dealt a private number $s_1, s_2 \sim$ Uniform$[0, \ldots, N-1]$. $\mathcal{A} = \{\text{Pass}, 2^0, \ldots, 2^k\}$ is an ordered set. The game alternates between P1 and P2, and P1 bids first. The bidding sequence is strictly increasing. The game ends if either player passes, and $r = 2^k$ if $s_1 + s_2 \geq 2^k$ where $k$ is the latest bid. Otherwise the contract fails and $r = 0$.*

**Definition 3** (2-Suit Mini-Bridge of size $N$). *P1 and P2 each dealt a private number $s_1, s_2 \sim$ Uniform$[0, 1, \ldots, N]$. $\mathcal{A} = \{\text{Pass}, 1\heartsuit, 1\spadesuit, 2\heartsuit, \ldots N\heartsuit, N\spadesuit\}$ is an ordered set. The game progresses as in Def. 2. Except for the first round, the game ends if either player passes. If $k\spadesuit$ is the last bid and $s_1 + s_2 \geq N + k$, or if $k\heartsuit$ is the last bid and $s_1 + s_2 \leq N - k$, then $r = 2^{k-1}$, otherwise the contract fails ($r = -1$). For pass out situation* (Pass, Pass), $r = 0$.

The communication game (Def. 1) can be perfectly solved to reach a joint reward of 1 with arbitrary binary encoding of $s_1$. However, there exists many local solutions where P1 and P2 agree on a subset of $s_1$ but have no consensus on the meaning of a new $L$-bit signal. In this case, a unilateral approach cannot establish such a consensus. The other two games are harder. In Simple Bidding (Def. 2), available actions are on the order of $\log(N)$, requiring P1 and P2 to efficiently communicate. The

Table 1: Average reward of multiple tabular games after optimizing policies using various approaches. Both CFR [44] and CFR1k+JPS repeats with 1k different seeds. BAD [16] runs 50 times. The trunk policy network of BAD uses 2 Fully Connected layers with 80 hidden units. Actor-Critic run 10 times. The super script $*$ means the method obtains the best known solution in *one* of its trials. We omit all standard deviations of the mean values since they are $\sim 10^{-2}$.

| | Comm (Def. 1) | | | | Mini-Hanabi | Simple Bidding (Def. 2) | | | 2SuitBridge (Def. 3) | | |
|---|---|---|---|---|---|---|---|---|---|---|---|
| | $L=3$ | $L=5$ | $L=6$ | $L=7$ | [16] | $N=4$ | $N=8$ | $N=16$ | $N=3$ | $N=4$ | $N=5$ |
| CFR1k [44] | 0.89* | 0.85 | 0.85 | 0.85 | 9.11* | 2.18* | 4.96* | 10.47 | 1.01* | 1.62* | 2.60 |
| CFR1k+JPS | **1.00*** | **1.00*** | **1.00*** | **1.00*** | **9.50*** | 2.20* | **5.00*** | **10.56*** | **1.07*** | **1.71*** | **2.74*** |
| A2C [27] | 0.60* | 0.57 | 0.51 | 0.02 | 8.20* | 2.19 | 4.79 | 9.97 | 0.66 | 1.03 | 1.71 |
| BAD [16] | **1.00*** | 0.88 | 0.50 | 0.29 | 9.47* | **2.23*** | 4.99* | 9.81 | 0.53 | 0.98 | 1.31 |
| **Best Known** | 1.00 | 1.00 | 1.00 | 1.00 | 10 | 2.25 | 5.06 | 10.75 | 1.13 | 1.84 | 2.89 |
| #States | 633 | 34785 | 270273 | 2129793 | 53 | 241 | 1985 | 16129 | 4081 | 25576 | 147421 |
| #Infosets | 129 | 2049 | 8193 | 32769 | 45 | 61 | 249 | 1009 | 1021 | 5116 | 24571 |

Mini-Bridge (Def. 3) mimics the bidding phase of Contract Bridge: since bids can only increase, both players need to strike a balance between reaching highest possible contract (for highest rewards) and avoiding overbids that lead to negative rewards. In this situation, forming a convention requires a joint policy improvement for both players.

For SimpleBidding ($N = 16$), MiniBridge ($N = 4, 5$), we run Alg. 1 with a search depth $D = 3$. For other games, we use maximal depth, i.e., from the starting infosets to the terminals. Note this does not involve all infosets, since at each depth only one active infoset exists. JPS never worsens the policy so we use its last solution. For A2C and BAD, we take the best model over 100 epoch (each epoch contains 1000 minibatch updates). Both A2C and BAD use a network to learn the policy, while CFR and JPS are tabular approaches. To avoid convergence issue, we report CFR performance after purifying CFR's resulting policy. The raw CFR performance before purification is slightly lower.

As shown in Tbl. 1, JPS consistently improves existing solutions in multiple games, in particular for complicated IGs (e.g. 2-Suit Mini-Bridge). See Appendix C for a good solution found by JPS in 2-suited Bridge. BAD [16] does well for simple games but lags behind in more complicated IGs.

We also tried different combinations between JPS and other solvers. Except for Comm (Def. 1) that JPS always gets 1.0, uniform random+JPS converges to local minima that CFR is immune to, and under-performs CFR1k+JPS. Combining JPS with more CFR iterations (CFR10k) doesn't improve performance. Compared to CFR1k+JPS, BAD+JPS is worse (10.47 vs 10.56 for $N = 16$) in Simple Bidding but *better* (1.12/1.71/2.77 vs 1.07/1.71/2.74 for $N = 3/4/5$) in 2-Suit Mini-Bridge. Note that this is quite surprising since the original solutions obtained from BAD are not great but JPS can boost them substantially. We leave these interesting interplays between methods for future study.

**Correctness of Theorem 1 and runtime speed**. Experiments show that the game value difference $\bar{v}^{\sigma'} - \bar{v}^{\sigma}$ from Theorem 1 always coincides with naive computation, with much faster speed. We have compared JPS with brute-force search. For example, for each iteration in Simple Bidding (Def. 2), for $N = 8$, JPS takes $\sim 1$s while brute-force takes $\sim 4$s (4x); for $N = 16$ and $d = 3$, JPS takes $\sim 20$s while brute-force takes $\sim 260$s (13x). For communication game (Def. 1), JPS enjoys a speedup of 3x for $L = 4$. For 2-Suit Mini-Bridge of $N = 4$, it achieves up to 30x.

## 7 Application to Contract Bridge Bidding

In this section, we apply the online version of JPS (Sec. 5.2) to the bidding phase of Contract Bridge (a 4-player game, 2 in each team), to improve collaboration between teammates from a strong baseline model. Note that we insert JPS in the general self-play framework to improve collaboration between teammates and thus from JPS's point of view, it is still a fully collaborative IG with fixed opponents. Unlike [43] that only models 2-player collaborative bidding, our baseline and final model are for full Bridge Bidding. Note that since Bridge is not a pure collaborative games and we apply an online version of JPS, the guarantees of Theorem. 2 is lost, while empirically it performs well.

**A Crash Course of Bridge Bidding**. The bidding phase of Contract Bridge is like Mini-Bridge (Def. 3) but with a much larger state space (each player now holds a hand with 13 cards from 4 suits). Unlike Mini-Bridge, a player has both her teammate and competitors, making it more than a full-collaborative IG. Therefore, multiple trade-offs needs to be considered. Human handcrafted conventions to signal private hands, called *bidding systems*. For example, opening bid $2\heartsuit$ used to signal a very strong hand with hearts historically, but now signals a weak hand with long hearts. Its current usage blocks opponents from getting their best contract, which happens more frequently than its previous usage (to build a strong heart contract). Please see Appendix A for more details.

**Evaluation Metric.** We adopt *duplicate bridge* tournament format: each board (hands of all 4 players) is played twice, where a specific team sits North-South in one game (called open table), and East-West in another (called close table). The final reward is the difference of the results of two tables. This reduces the impact of card dealing randomness and can better evaluate the strength of an agent.

We use IMPs (International Matching Point) per board, or *IMPs/b*, to measure the strength difference between two Bridge bidding agents. See Appendix A for detailed definition. Intuitively, *IMPs/b* is the normalized score difference between open and close table in duplicate Bridge, ranging from $-24$ to $+24$. In Compute Bridge, a margin of $+0.1$ *IMPs/b* is considered significant [32]. In a Bridge tournament, a forfeit in a game counts as $-3$ *IMPs/b*. The difference between a top professional team and an advanced amateur team is about 1.5 *IMPs/b*.

**Reward**. We focus on the bidding part of the bridge game and replace the playing phase with Double Dummy Solver (DDS) [19], which computes the maximum tricks each team can get in playing, if all actions are optimal given full information. While this is not how humans plays and in some situations the maximum tricks can only be achieved with full-information, DDS is shown to be a good approximate to human expert plays [32]. Therefore, after bidding we skip the playing phase and directly compute *IMPs/b* from the two tables, each evaluated by DDS, as the only sparse reward.

Note that Commercial software like Wbridge5, however, are not optimized to play under the DDS setting, and we acknowledge that the comparison with Wbridge5 is slightly unfair. We leave end-to-end evaluation including the playing phase as future work.

**Dataset**. We generate a training set of 2.5 million hands, drawn from uniform distribution on permutations of 52 cards. We pre-compute their DDS results. The evaluation dataset contains 50k such hands. Both datasets will be open sourced for the community and future work.

**Baselines**. We use `baseline16` [43], `baseline19` [32] and `baseline` [18] as our baselines, all are neural network based methods. See Appendix B for details of each baseline.

## 7.1 Network and Training

We use the same network architecture as `baseline`, which is also similar to `baseline19`. As show in Fig. 2, the network consists of an initial fully connected layer, then 4 fully connected layer with skip connections added every 2 layers to get a latent representation. We use 200 neurons at each hidden layer, so it is much smaller (about 1/70 parameter size of `baseline19`).

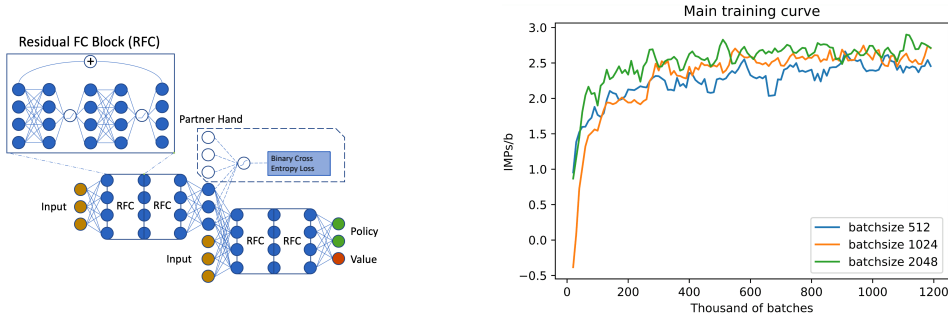

Figure 2: Left: Network Architecture. Supervision from partner's hand is unused in the main results, and is used in the ablation studies. Right: Smoothed training curves for different batchsizes.

**Input Representation**. For network input, we use the same encoding as `baseline`. This includes 13 private cards, bidding sequence so far and other signals like vulnerability and legal actions. Please check Appendix D for details. The encoding is general without much domain-specific information. In contrast, `baseline19` presents a novel bidding history representation using positions in the maximal possible bidding sequence, which is highly specific to Contract Bridge.

## 7.2 A Strong Baseline Model

We train a strong baseline model for 4-player Bridge Bidding with A2C [27] with a replay buffer, importance ratio clipping and self-play. During training we run 2000 games in parallel, use batch size of 1024, an entropy ratio of 0.01 and with no discount factor. See Appendix E for details.

Fig. 2 shows example training curves against `baseline16`. We significantly outperform `baseline16` by a huge margin of +2.99 *IMPs/b*. This is partially because `baseline16` cannot adapt well to competitive bidding setting. Also it can also only handle a fixed length of bids. We have performed an extensive ablation study to find the best combination of common tricks used in

DRL. Surprisingly, some of them believed to be effective in games, e.g., explicit belief modeling, have little impact for Bridge bidding, demonstrating that unilateral improvement of agent's policy is not sufficient. See Appendix F for a detailed ablation study.

Table 2: Fine-tuning RL pre-trained model with search applied on $1\%$ games or moves unless otherwise stated. Performance in *IMPs/b*. 10 baselines are other independently trained actor-critic baselines.

|  | vs. baseline | vs. 10 baselines |
| --- | --- | --- |
| non-search | 0.20 | $0.27 \pm 0.13$ |
| 1-search | 0.46 | $0.37 \pm 0.11$ |
| JPS (1%) | **0.71** | $0.47 \pm 0.11$ |
| JPS (5%) | 0.70 | $\mathbf{0.66 \pm 0.11}$ |
| JPS (10%) | 0.44 | $0.39 \pm 0.11$ |

### 7.3   JPSBid: Improving strong baseline models with JPS

We then use JPS to further improve the strong baseline model. Similar to Sec. 6, JPS uses a search depth of $D = 3$: the current player's (P1) turn, the opponent's turn and the partner's (P2) turn. We only jointly update the policy of P1 and P2, assuming the opponent plays the current policy $\sigma$. After the P1's turn, we rollout 5 times to sample opponent's actions under $\sigma$. After P2's turn, we rollout 5 times following $\sigma$ to get an estimate of $v^{\sigma}(h)$. Therefore, for each initial state $h_0$, we run $5 \times 5$ rollouts for each combination of policy candidates of P1 and P2. Only a small fraction (e.g., $5\%$) of the games stopped at some game state and run the search procedure above. Other games just follow the current policy $\sigma$ to generate trajectories, which are sent to the replay buffer to stabilize training. A game thread works on one of the two modes decided by rolling a dice.

We also try a baseline `1-search` which only improve P1's policy (i.e., $D = 1$). And `non-search` baseline is just to reload the baseline model and continue A2C training.

From the training, we pick the best model according to its *IMPs/b* against the baseline, and compare with 10 other baseline models independently trained with A2C with different random seeds. They give comparable performance against `baseline16`.

Tbl. 2 shows a clear difference among `non-search`, `1-search` and JPS, in particular in their transfer performance against independent baselines. JPS yields much better performance (+0.66 *IMPs/b* against 10 independent baselines). We can observe that `1-search` is slightly better than `non-search`. With JPS, the performance gains significantly.

**Percentage of search**.  Interestingly, performing search in too many games is not only computationally expensive, but also leads to model overfitting, since the trajectories in the replay buffer are infrequently updated. We found that 5% search performs best against independent baselines.

**Against WBridge5**.  We train our bot with JPS for 14 days and play 1000 games between our bot and WBridge5, a software winning multiple world champion in 2005, 2007, 2008 and 2016. The 1000 games are separately generated, independent of training and evaluation set. We outperform by a margin of +0.63 *IMPs/b* with a standard error of 0.22 *IMPs/b*. This translates to 99.8% probability of winning in a standard match. This also surpasses the previous SoTAs `baseline`[18] (+0.41 *IMPs/b* evaluated on 64 games only), and `baseline19` (+0.25 *IMPs/b*). Details in Appendix H.

Note that we are fully aware of the potential unfairness of comparing with WBridge5 only at Bridge bidding phase. This includes that **(1)** WBridge5 conforms to human convention but JPS can be creative, **(2)** WBridge5 optimizes for the results of real Bridge playing rather than double-dummy scores (DDS) that assumes full information during playing, which is obviously very different from how humans play the game. In this paper, to verify our bot, we choose to evaluate against WBridge5, which is an independent baseline tested extensively with both AI and human players. A formal address of these issues requires substantial works and is left for future work.

**Visualzation of Learned models.**  Our learned model is visualized to demonstrate its interesting behaviors (e.g., an aggressive opening table). We leave detailed discussion in the Appendix I.

## 8   Conclusion and Future Work

In this work, we propose JPS, a general optimization technique to jointly optimize policy for collaborative agents in imperfect information game (IG) efficiently. On simple collaborative games, tabular JPS improves existing approaches by a decent margin. Applying online JPS in competitive Bridge Bidding yields SoTA agent, outperforming previous works by a large margin (+0.63 *IMPs/b*) with a $70\times$ smaller model under Double-Dummy evaluation. Future works include applying JPS to other collaborative IGs with various advanced search techniques and studying sub-optimal equilibria.

## 9 Broader Impact

This work has the following potential positive impact in the society:

- JPS proposes a general formulation and can be applied to multi-agent pure collaborative games (or team collaboration compnents in multi-agent games) beyond the simple games and Contract Bridge we demonstrate in the paper;
- JPS can potentially encourage more efficient collaboration between agents and between agents and humans. It might suggest novel coordination patterns, helping jump out of existing (but sub-optimal) social convention.

We do not foresee negative societal consequences from JPS.

## Footnotes

[1]In the bidding phase, asides the current player, each of the other 3 players can hold $6.35 \times 10^{11}$ unique hands and there are $10^{47}$ possible bidding sequences. Unlike hint games like Hanabi [2], public actions in Bridge (e.g. bid) do not have pre-defined meaning and does not decrease the uncertainty when game progresses.

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
