[Supplementary Material]

# A The Contract Bridge Game

The game of Contract Bridge is played with a standard 52-card deck (4 suits, ♠, ♡, ◇ and ♣, with 13 cards in each suit) and 4 players (North, East, South, West). North-South and East-West are two competitive teams. Each player is dealt with 13 cards.

There are two phases during the game, namely **bidding** and **playing**. After the game, **scoring** is done based on the won tricks in the playing phase and whether it matches with the contract made in the bidding phase. An example of contract bridge bidding and playing in shown in Fig. 3.

Figure 3: **(a)** A bidding example. North-South prevail and will declare the contract 4♠. During the bidding, assuming natural bidding system, the bid 1♠, 2♣, 4♣ and 4♠ are natural bids, which shows lengths in the nominated suit. The bid 3♣ is an artificial bid, which shows a good hand with ♠ support for partner, and shows nothing about the ♣ suit. To make the contract, North-South needs to take 10 tricks during the playing phase. **(b)** A playing example. Currently shown is the 2nd round of the playing phase. The dummy's card is visible to all players, and controlled by his partner, declarer. In the current round North player wins with ♣K, and will lead the next round.

**Bidding phase**. During the bidding phase, each player takes turns to bid from 38 available actions. The sequence of bids form an auction. There are 35 contract bids, which consists a level and a strain, ranging from an ordered set $\{1♣, 1d, 1♡, 1♠, 1NT, 2♣, ..7NT\}$ where NT stands for No-Trump. The level determines the number of tricks needed to make the contract, and the strain determines the trump suit if the player wins the contract. Each contract bid must be either higher in level or higher in strain than the previous contract bids.

There are also 3 special bids. **Pass (P)** is always available when a player is not willing to make a contract bid. Three consecutive passes will terminate the auction, and the last contract bid becomes the final contract, with their side winning the contract. If the auction has 4 Passes, then the game ends with reward 0 and restarts. **Double (X)** can be used when either opponent has made a contract bid. It will increase both the contract score, if the declarer makes the contract, and the penalty score for not making the contract. Originally this is used when a player has high confidence that opponent's contract cannot be made, but it can also be used to communicate information. Finally, **Redouble (XX)** can be used by the declaring team to further amplify the risk and/or reward of a contract, if the contract is doubled. Similarly, this bid can also be used to convey other information.

**Playing phase**. After the bidding phase is over, the contract is determined, and the owner of the final contract is the declarer. His partner becomes dummy. The other partnership is the defending side. During the playing phase, there are 13 rounds and each rounds the player plays a card. The first round starts with the defending side, and then dummy immediately lays down his cards, and the declarer can control the cards of both himself and dummy. The trump suit is designated by the strain of the final contract (Or None if the strain is NT). Each round, every player has to follow suit. If a player is out of a certain suit, he can play a trump card to beat it. Discarding other suits is always losing in this round. The player who played the highest ranked card (or play a trump) wins a trick, and will play first in the next round. The required number of tricks for the declarer's team to make the contract is contract level + 6 (e.g., 1♠ means that 7 tricks are needed). At the end of the game, if the declaring side wins enough tricks, they make the contract. Tricks in addition to the required tricks are called over-tricks. If they fail to make the contract, the tricks short are called under-tricks.

**Scoring**. if the contract is made, the declaring side will receive contract score as a reward, plus small bonuses for over-tricks. Otherwise they will receive negative score determined by under-tricks. Contracts below 4♡ (except 3NT) are called **part score contracts**, with relatively low contract scores. Contracts 4♡ and higher, along with 3NT, are called **game contracts** with a large bonus

score. Finally, Contract with level 6 and 7 are called **small slams** and **grand slams** respectively, each with a huge bonus score if made. To introduce more variance, *vulnerability* is randomly assigned to each board to increase bonuses/penalties for failed contracts.

In *duplicate bridge*, two tables are played with exact the same full-hands. Two players from one team play North-South in one table, and two other players from the same team play East-West in the other table. After raw scores are assigned to each table, the difference is converted to IMPs scale [2] in a tournament match setting, which is roughly proportional to the square root of the raw score, and ranges from 0 to 24.

## B    Baselines

Previous works tried applying DRL on Bridge bidding.

`baseline16` [43] uses DRL to train a bidding model in the collaborative (2-player) setting. It proposes Penetrative Bellman's Equation (PBE) to make the Q-function updates more efficient. The limitation is that PBE can only handle fixed number of bids, which are not realistic in a normal bridge game setting. As suggested by the authors, we modify their pre-trained model to bid competitively (i.e., 4 players), by bidding PASS if the cost of all bids are greater than 0.2. We implement this and further fix its weakness that the model sometimes behaves randomly in a competitive setting if the scenario can never occur in a collaborative setting. We benchmark against them at each episode.

`baseline19` [32] proposes two networks, Estimation Neural Network (ENN) and Policy Neural Network (PNN) to train a competitive bridge model. ENN is first trained supervisedly from human expert data, and PNN is then learned based on ENN. After learning PNN and ENN from human expert data, the two networks are further trained jointly through reinforcement learning and selfplay. PBE claims to be better than Wbridge5 in the collaborative (2-player) setting, while PNN and ENN outperforms Wbridge5 in the competitive (4-player) setting. We could not fully reproduce its results so we cannot directly compare against `baseline19`. However, since both our approach and `baseline19` have compared against WBridge5, we indirectly compare them.

`baseline` [18] is trained with large-scale A2C, similar to our approach, but without the JPS improvement. Furthermore, when evaluating with WBridge5, only 64 games are used. We indirectly compare them thought performance against Wbridge5 on 1000 games.

Policy Belief Learning (PBL) [41] proposes to alternately train between policy learning and belief learning over the whole self-play process. Like `baseline16`, the Bridge agent obtained from PBL only works in collaborative setting.

## C    Trained Policy on 2-Suit MiniBridge

We show a learned policy with JPS on 2-suit Mini-Bridge with $N = 4$ in Tbl. 3 , which received the maximal score (1.84). We find that the learned policy did well to bid optimal contracts in most scenarios. On the anti-diagonal (0/4 and 4/0 case in the table), no contracts can be made, but in order to explore possible high reward contracts, the agent have to bid, leading to some overbid contracts.

Table 3: Trained Policy on 2-Suit Mini-Bridge. Rows are the number of ♡s of Player 1 and Columns are the number of ♡s of Player 2.

|   | 0 | 1 | 2 | 3 | 4 |
|---|---|---|---|---|---|
| 0 | 1♠-2♠-3♠-4♠-P | 1♠-2♠-3♠-P | 1♠-2♡-2♠-P | 1♠-P | 1♠-2♡-2♠-4♡-4♠-P |
| 1 | P-2♠-3♠-P | P-1♠-2♠-P | P-P | P-P | P-1♡-P |
| 2 | P-2♠-P | P-1♠-P | P-P | P-P | P-1♡-2♡-P |
| 3 | 1♡-1♠-P | 1♡-P | 1♡-P | 1♡-2♡-P | 1♡-3♡-P |
| 4 | 1♡-1♠-4♡-4♠-P | 1♡-P | 1♡-P | 1♡-2♡-3♡-P | 1♡-3♡-3♠-4♡-P |

| Own Cards | Bid history, multi-hot of contract bids | | | | | Vulner-ability | Available Action |
|---|---|---|---|---|---|---|---|
| | Own | Partner | Left Opp | Right Opp | Double Indicator | | |
| 52 | 35 | 35 | 35 | 35 | 35 | 2 | 38 |

| W | N | E | S |
|---|---|---|---|
| | 1♣ | X | 1♥ |
| 1♠ | 2♣ | P | ? |

Figure 4: Input representation. With the decision point shown in the example, South will mark the following bits in the bidding history encoding: 1♡ in "Own" segment, 1♣ and 2♣ in "Partner" segment, 1♠ in "Left Opp" segment, and 1♣ in "Double Indicator" segment.

## D  Input Representation

We encode the state of a bridge game to a 267 bit vector as shown in Fig. 4. The first 52 bits indicate that if the current player holds a specific card. The next 175 bits encodes the bidding history, which consists of 5 segments of 35 bits each. These 35 bit segments correspond to 35 contract bids. The first segment indicates if the current player has made a corresponding bid in the bidding history. Similarly, the next 3 segments encodes the contract bid history of the current player's partner, left opponent and right opponent. The last segment indicates that if a corresponding contract bid has been doubled or redoubled. Since the bidding sequence can only be non-decreasing, the order of these bids are implicitly conveyed. The next 2 bits encode the current vulnerability of the game, corresponding to the vulnerability of North-South and East-West respectively. Finally, the last 38 bits indicates whether an action is legal, given the current bidding history.

Note that the representation is *imperfect recall*: from the representation the network only knows some bid is doubled by the opponent team, but doesn't know which opponent doubles that bid. We found that it doesn't make a huge difference in terms of final performance.

## E  Training Details

We train the model using Adam with a learning rate of 1e-4. During training we use multinominal exploration to get the action from a policy distribution, and during evaluation we pick the greedy action from the model. We also implement a replay buffer of size 800k, and 80k burn in frames to initialize the replay buffer.

**RL Method and Platform Implementation**. We use selfplay on random data to train our baseline models. The baseline model is trained with A2C [27] with replay buffer, off-policy importance ratio correction/capping and self-play, using ReLA platform[3]. ReLA is an improved version of ELF framework [38] using PyTorch C++ interface (i.e., TorchScript). ReLA supports off-policy training with efficient replay buffer. The game logic of Contract Bridge as well as feature extraction steps are implemented in C++ and runs in parallel to make the training fast. Each player can call different models directly in C++ and leaves action trajectories to the common replay buffer, making it suitable for multi-agent setting. The training is thus conducted in a separated Python thread by sampling batches from the replay buffer, and update models accordingly. The updated model is sent back to the C++ side for further self-play, once every **Sync Frequency** minibatches.

We improve the open source version of ReLA to support dynamic batching in rollouts and search. Unlike ELF that uses thousands of threads for simulation, we now put multiple environments in a single C++ thread to reduce the context-switch cost, while the dynamic batching mechanism can still batch over these environments, using a mechanism provided by **std::promise** and **std::future**. This gives $\sim$ 11x speedup compared to a version without batching. The platform is efficient and can evaluate 50k games using pre-trained models in less than a minute on a single GPU. During training, to fill in a replay buffer of 80k transitions, it takes 2.5 seconds if all agents play with current policy, and $\sim$ 1 minute if all agents use JPS in 100% of its actions. The whole training process takes roughly 2 days on 2-GPUs. We also try training a 14-day version of JPS model.

Table 4: Performance Comparison. The table compares performance when giving different weights to the belief loss and other hyper-parameters such as number of RFC blocks in the network and actor sync frequency.

| Ratio r | imps ± std | Num Blocks | imps ± std | Sync Frequency | imps ± std |
|---------|------------|------------|------------|----------------|------------|
| 0 | **2.99 ± 0.04** | 2 | 2.97 ± 0.05 | 1 | 2.89 ± 0.13 |
| 0.001 | 2.86 ± 0.18 | 4 | **2.99 ± 0.04** | 6 | 2.92 ± 0.16 |
| 0.01 | 2.77 ± 0.22 | 10 | 2.94 ± 0.15 | 12 | 2.94 ± 0.14 |
| 0.1 | 2.53 ± 0.27 | 20 | **2.99 ± 0.06** | 50 | **2.99 ± 0.04** |

# F    Ablation Studies

## F.1    A2C baseline

We perform extensive ablation studies for A2C self-play models, summarized in Tbl. 4. Our attempts to improve its performance by applying existing methods and tuning hyper-parameters yield negative results.

One example is explicit **belief modeling** (e.g., with auxiliary loss [32] or alternating training stages [41]), we found that it doesn't help much in Bridge bidding. We use $L = rL_{belief} + L_{A2C}$ as the loss, where $r$ is a hyper-parameter to control the weight on the auxiliary task, As shown in Table 4, when $r = 0$, the model reaches the best performance and the performance decreases as $r$ increase. This shows that it might be hard to move out of local minima with auxiliary loss, compared to search-based approaches. Adding **more blocks** of FC network cannot further improve its performance, showing that model capacity is not the bottleneck. The performance is similar when the **sync frequency** is large enough.

## F.2    Joint Policy Exploration

It is possible that Joint Policy Search (JPS) works just because it encourages joint exploration. To distinguish the two effects, we also run another baseline in which the agent and its partner explore new actions simultaneously but randomly. We find that this hurts the performance, compared to independent exploration. This shows that optimizing the policy of the current player and its partner jointly given the current policy is important for model improvement.

Table 5: Joint Exploration hurts the performance.

| Joint Random Exploration Ratio | imps ± std |
|--------------------------------|------------|
| 0 | **2.99 ± 0.04** |
| 0.001 | 2.43 ± 0.20 |
| 0.01 | 2.37 ± 0.31 |

# G    Details of competing with WBridge5 and additional results

**Experimental settings.** We compare with WBridge5, which is an award-winning close-sourced free software[4]. Since it can only run on Microsoft Windows, we implement a UI interface to mimic keyboard and mouse moves to play against WBridge5. Our model controls one player and its partner, while WBridge5 controls the other two players (in summary, 2 **JPSBid** are teamed up against 2 WBridge5 agents). Note that the two players cannot see each other's private information, while their model architecture and parameters are shared. For each model, we use 1000 different board situations and compare its mean estimate (in *IMPs/b*) and standard error of the mean estimate. These 1000 board situations are generated as a separate test set from the training and validation set.

Table 6 shows the performance. Interestingly, while $5\%$ **JPSBid** gives good performance when comparing against 10 independent baselines, it is slightly worse than $1\%$ version when competing with WBridge5. This is likely due to insufficient self-play data produced by expensive rollout operations that involve search.

Table 6: Performance against WBridge5.

| | Vs. WBridge5 (*IMPs/b*) |
|---|---|
| A2C baseline | $0.29 \pm 0.22$ |
| 1% search, **JPSBid** (2 days) | $0.44 \pm 0.21$ |
| 1% search, **JPSBid** (14 days) | $\mathbf{0.63 \pm 0.22}$ |
| 5% search, **JPSBid** (2 days) | $0.38 \pm 0.20$ |

Figure 5: Bidding length histogram.

## H   Statistics of learned models

### H.1   Bidding Statistics

It is interesting to visualize what the model has learned, and understand some rationales behind the learned conventions. In Fig. 5 and Tbl. 7, we show the bidding length distribution and frequency of each bid used, as well as the distribution of final contracts. We can see that typically agents exchange 6-15 rounds of information to reach the final contract. The agent uses low level bids more frequently and puts an emphasis on ♡ and ♠ contracts. The final contract is mostly part scores and game contracts, particularly 3NT, 4♡ and 4♠. This is because part scores and game contracts are optimal based on DDS for 87% of hands[5]. As a result, the model will optimize to reach these contracts.

Table 7: Most frequent bids and final contracts.

| Bids | Frequency | Final Contracts | Frequency |
|---|---|---|---|
| P | 57.31% | 2♡ | 8.07% |
| 1♣ | 3.74% | 2♠ | 7.83% |
| 1♠ | 3.23% | 1NT | 7.71% |
| X | 3.16% | 3d | 7.34% |
| 2♡ | 3.10% | 3NT | 6.58% |
| 2♠ | 2.84% | 4♡ | 5.90% |
| 1NT | 2.84% | 4♠ | 5.23% |

### H.2   Opening Table

There are two mainstream bidding systems used by human experts. One is called *natural*, where opening and subsequent bids usually shows length in the nominated suit, e.g. the opening bid 1♡ usually shows 5 or more ♡ with a decent strength. The other is called *precision*, which heavily relies on relays of bids to partition the state space into meaningful chunks, either in suit lengths or hand strengths, so that the partner knows the distribution of the private card better. For example, an opening bid of 1♣ usually shows 16 or more High Card Points (HCP)[6], and a subsequent 1♡ can show 5 or more ♠. To further understand the bidding system the model learns, it is interesting to establish an opening table of the model, defined by the meaning of each opening bid. We select one of the best models, and check the length of each suit and HCP associated with each opening bid. From the opening table, it appears that the model learns a semi-natural bidding system with very aggressive openings (i.e., high bid even with a weak private hand).

Table 8: Opening table comparisons. "bal" is abbreviation for a balanced distribution for each suit.

| Opening bids | Ours | SAYC |
|---|---|---|
| 1♣ | 10+ HCP | 12+ HCP, 3+♣ |
| 1d | 8-18 HCP, <4 ♡, <4 ♠ | 12+ HCP, 3+d |
| 1♡ | 4-16 HCP, 4-6♡ | 12+ HCP, 5+♡ |
| 1♠ | 4-16 HCP, 4-6♠ | 12+ HCP, 5+♠ |
| 1NT | 12-17 HCP, bal | 15-17 HCP, bal |
| 2♣ | 6-13 HCP, 5+♣ | 22+ HCP |
| 2d | 6-13 HCP, 5+d | 5-11 HCP, 6+d |
| 2♡ | 8-15 HCP, 5+♡ | 5-11 HCP, 6+♡ |
| 2♠ | 8-15 HCP, 5+♠ | 5-11 HCP, 6+♠ |

# I  Proofs

## I.1  Lemma 1

*Proof.* Let $I = I(h)$, since $\sigma(I, a) = \sigma'(I, a)$, we have:

$$\sum_{a \in A(I)} \boldsymbol{c}^{\sigma,\sigma'}(ha) := \sum_{a \in A(I)} (\pi^{\sigma'}(ha) - \pi^{\sigma}(ha))\boldsymbol{v}^{\sigma}(ha) \tag{8}$$

$$= (\pi^{\sigma'}(h) - \pi^{\sigma}(h)) \sum_{a \in A(I)} \sigma(I, a)\boldsymbol{v}^{\sigma}(ha) \tag{9}$$

$$= (\pi^{\sigma'}(h) - \pi^{\sigma}(h))\boldsymbol{v}^{\sigma}(h) \tag{10}$$

$$= \boldsymbol{c}^{\sigma,\sigma'}(h) \tag{11}$$

Therefore, $\boldsymbol{\rho}^{\sigma,\sigma'}(h) := -\boldsymbol{c}^{\sigma,\sigma'}(h) + \sum_{a \in A(I)} \boldsymbol{c}^{\sigma,\sigma'}(ha) = \boldsymbol{0}$. ☐

## I.2  Subtree decomposition

**Lemma 3.** *For a perfect information subtree rooted at $h_0$, we have:*

$$\pi^{\sigma'}(\boldsymbol{v}^{\sigma'} - \boldsymbol{v}^{\sigma})|_{h_0} = \sum_{h_0 \sqsubseteq h \notin Z} \boldsymbol{\rho}^{\sigma,\sigma'}(h) \tag{12}$$

*Proof.* First by definition, we have for any policy $\sigma'$:

$$\boldsymbol{v}^{\sigma'}(h_0) = \sum_{z \in Z} \pi^{\sigma'}(z|h_0)\boldsymbol{v}(z) \tag{13}$$

where $\pi^{\sigma'}(z|h_0) := \pi^{\sigma'}(z)/\pi^{\sigma'}(h_0)$ is the *conditional* reachability from $h_0$ to $z$ under policy $\sigma'$. Note that $\boldsymbol{v}(z)$ doesn't depend on policy $\sigma'$ since $z$ is a terminal node.

We now consider each terminal state $z$. Consider a path from game start $h_0$ to $z$: $[h_0, h_1, \ldots, z]$. With telescoping sum, we could write:

$$\pi^{\sigma'}(z|h_0)\boldsymbol{v}(z) = \pi^{\sigma'}(h_0, z|h_0)\boldsymbol{v}^{\sigma}(h_0) + \sum_{h:\, h \sqsubseteq z, ha \sqsubseteq z} \pi^{\sigma'}(ha, z|h_0)\boldsymbol{v}^{\sigma}(ha) - \pi^{\sigma'}(h, z|h_0)\boldsymbol{v}^{\sigma}(h) \tag{14}$$

where $\pi^{\sigma'}(h, z|h_0)$ is the joint probability that we reach $z$ through $h$, starting from $h_0$. Now we sum over all possible terminals $z$ that are descendants of $h_0$ (i.e., $h_0 \sqsubseteq z$). Because of the following,

- From Eqn. 13, the left-hand side is $\boldsymbol{v}^{\sigma'}(h_0)$;

- For the right-hand side, note that $\sum_{z:\, h \sqsubseteq z} \pi^{\sigma'}(h, z|h_0) = \pi^{\sigma'}(h|h_0)$. Intuitively, this means that the reachability of $h$ is the summation of all reachabilities of the terminal nodes $z$ that are the consequence of $h$.

we have:

$$\boldsymbol{v}^{\sigma'}(h_0) = \pi^{\sigma'}(h_0|h_0)\boldsymbol{v}^{\sigma}(h_0) + \sum_{h_0 \sqsubseteq h \notin Z} \sum_{a \in A(h)} \pi^{\sigma'}(ha|h_0)\boldsymbol{v}^{\sigma}(ha) - \pi^{\sigma'}(h|h_0)\boldsymbol{v}^{\sigma}(h) \quad (15)$$

Notice that $\pi^{\sigma'}(h_0|h_0) = 1$ and if we multiple both side by $\pi^{\sigma'}(h_0)$, we have:

$$\pi^{\sigma'}(\boldsymbol{v}^{\sigma'} - \boldsymbol{v}^{\sigma})|_{h_0} = \sum_{h_0 \sqsubseteq h \notin Z} \sum_{a \in A(h)} \pi^{\sigma'}(ha)\boldsymbol{v}^{\sigma}(ha) - \pi^{\sigma'}(h)\boldsymbol{v}^{\sigma}(h) \quad (16)$$

$$= \sum_{h_0 \sqsubseteq h \notin Z} \pi^{\sigma'}(h) \sum_{a \in A(h)} \sigma'(I(h), a)\boldsymbol{v}^{\sigma}(ha) - \boldsymbol{v}^{\sigma}(h) \quad (17)$$

$$= \sum_{h_0 \sqsubseteq h \notin Z} \boldsymbol{\rho}^{\sigma,\sigma'}(h) \quad (18)$$

This concludes the proof. □

## I.3   Lemma 2

*Proof.* Applying Lemma 3 and set $h_0$ to be the game start. Then $\pi^{\sigma'}(h_0) = 1$ and all $h$ are descendant of $h_0$ (i.e., $h_0 \sqsubseteq h$):

$$\bar{\boldsymbol{v}}^{\sigma'} - \bar{\boldsymbol{v}}^{\sigma} = \sum_{h \notin Z} \boldsymbol{\rho}^{\sigma,\sigma'}(h) \quad (19)$$

□

## I.4   Thm. 1

*Proof.* By Lemma 2, we have:

$$\bar{\boldsymbol{v}}^{\sigma'} - \bar{\boldsymbol{v}}^{\sigma} = \sum_{h \notin Z} \boldsymbol{\rho}^{\sigma,\sigma'}(h) = \sum_{I} \sum_{h \in I} \boldsymbol{\rho}^{\sigma,\sigma'}(h) \quad (20)$$

By Lemma 1, for all infoset set $I$ with $\sigma(I) = \sigma'(I)$, all its perfect information states $h \in I$ has $\boldsymbol{\rho}^{\sigma,\sigma'}(h) = \boldsymbol{0}$. The conclusion follows. □

## I.5   Thm. 2

*Proof.* According to Thm. 1, Alg. 1 computes $\bar{v}^{\sigma'} - \bar{v}^{\sigma}$ correctly for each policy proposal $\sigma'$ and returns the best $\sigma^*$. Therefore, we have

$$\bar{v}^{\sigma^*} - \bar{v}^{\sigma} = \max_{\sigma'} \bar{v}^{\sigma'} - \bar{v}^{\sigma} \geq 0 \quad (21)$$

□

## Footnotes

[2] https://www.acbl.org/learn_page/how-to-play-bridge/how-to-keep-score/duplicate/

[3]https://github.com/facebookresearch/rela

[4]http://www.wbridge5.com/

[5]https://lajollabridge.com/Articles/PartialGameSlamGrand.htm

[6]High Card Points is a heuristic to evaluate hand strength, which counts A=4, K=3, Q=2, J=1