[Reviews · NeurIPS 2020]

Review 1

Summary and Contributions: This paper looks at cooperative Imperfect Information games where players must jointly try new strategies to ensure the best equilibria are found. The paper introduces the idea of policy-change density, which allows for the impact of policy changes to be quantified as the sum of local policy-change densities across different information sets. An algorithm which improves policies using this idea is shown to guarantee that policy changes will only be improvements in the tabular case. Experimental validation of this algorithm is carried out in some small games, as well as in the bidding phase of 4-player contract bridge. The algorithm is shown to effectively improve upon policies generated by existing baseline algorithms.

Strengths: The notion of policy-change density are to my knowledge novel and the prospect of being able to evaluate policy changes with a local quantity has the potential of impacting work in these types of environments. The theory is clearly explained, and the experimental results are sound. This topic area is certainly relevant to the NeurIPS community.

Weaknesses: The only weaknesses of the work that I see are some limitations in the scope of the experimental results. The use of smaller domains was nice, but the lack of a complete comparison to another agent in the full contract bridge game (not just the bidding) is a little disappointing. I also feel that section 3.4 could use some extra space to elaborate and explain how these methods can be applied in the non-tabular case.

Correctness: The claims and methodology appear to be correct. The results could be strengthened by including full Bridge games or other domains, especially as the authors note that the comparison with Wbridge5 is "slightly unfair" (line 252).

Clarity: Yes, the paper is well written and clear. A few minor exceptions are noted in the additional feedback section.

Relation to Prior Work: Yes, this is discussion is adequate.

Reproducibility: Yes

Additional Feedback: Questions/Comments - There is a slight inconsistency between Equations (1) and (3), where in (1) you have A(I(h)) and in (3) you have A(h) - Line 142 - What is meant by the notation with a bar over the v? I don't see this introduced anywhere. This is a bit confusing, since your main theorem involves the difference between two overbar v quantities. It seems like this might be the value of the root node under the policy, but that is not explicitly stated anywhere. - It says at the beginning of Section 4 that JPS can be used to improve any existing strateg. It looks like you use the CFR1k strategy as a starting point for JPS. Do you experiment with using the other strategies (BAD and A2C) as starting points? How do the iterations of JPS compare from a time standpoint? In the last paragraph of Section 4 it says that JPS has a speedup when compared with brute-force, how does it compare with other possible strategies, or is brute-force the only thing that is directly comparable? - What happens if you start with a uniform random strategy and then use JPS? Does it converge to a good strategy at a similar rate to other algorithms (i.e. CFR)? - I didn't really understand what was meant by the "Percentage of Search" in section 5.3. How does this fit into Algorithm 1? - Can you evaluate your JPS bridge agent against the previous state-of-the-art [18] directly? Do we know that winning margins are transitive in Contract Bridge? Typos/Small Errors - Line 17 in abstract "test it to" should be "test it on" or "apply it to" - Line 58-59 is a little unclear. I think you mean "Using JPS to improve upon a strong A2C baseline agent, the resulting agent . . ." - Line 66, an "and" is required before "Monte Carlo . . . " - Line 74 ". . . in large-scale IIG that requires . . . " should be ". . . in large-scale IIGs that require . . ." given that you list multiple IIGs - Line 91 should be "the bidding phase" - footnote page 2 "asides" should be "aside from", and "does not" should be "do not" - Line 107 "coined" should be "coin" - Line 230 - The sentence beginning "Human handcrafted . . ." is incomplete. Did you mean "Humans handcrafted . . . "? - Line 308 - "Visualzation" should be "Visualization" ----------------------------- Post Author Rebuttal ----------------------------- I thank the author for their feedback and answers to questions.


Review 2

Summary and Contributions: The authors introduce a metric for evaluating policy changes locally in cooperative games, and use this to define an algorithm that searches in policy space for improved policies. This algorithm is presented in both exact tabular form and an approximate sampled form. The algorithm is applied to small games and to contract bridge bidding, with good results in both cases.

Strengths: The metric for policy change and the algorithm for search are both sound. Results on small cooperative games are good, but not perfect. The algorithm does well on bridge compared to previous work.

Weaknesses: The justification for the algorithm is intuitive, but there are no theoretical guarantees. It is easy to construct toy games on which this approach will not work. The paper would have benefited from a discussion of what properties a game would need to have for the algorithm to be effective, and also from a discussion of games on which the algorithm is known not to work.

Correctness: The evaluation vs WBridge5 is troublesome for various reasons, some of which are enumerated by the authors. One major additional reason is that bridge requires full disclosure (see Law 40 of the Laws of Contract Bridge, or https://www.acbl.org/conduct-and-ethics/active-ethics/#Disclosure). The opponents are entitled to know what hands are consistent with the actions taken. In computer tournament play, a human operator will be told what opposing actions mean and translate this into a description for WBridge5, to enable it to adapt its play accordingly. However, the authors do not make use of this facility, and so WBridge5 will assume that the opponent actions are in accordance with its own policy. This may cause WBridge5 to make incorrect inferences and to fail to adapt to the opponent policy. Although the agent suffers the same lack of information, it is reasonable to expect it to be more robust to differing opponent policies since it has been trained in the context of varying opponents. I don't think this problem can be reasonably addressed without significant additional work, and I don't think it disqualifies this paper from publication. However, I do think the claims of significant out-performance are unjustified. The evaluation against WBridge5 certainly demonstrates that the agent learns how to play the game effectively, but I don't think it's reasonable to claim more than that.

Clarity: The paper is clear, and well written, except for the obfuscated presentation of the policy change metric. However, the supplementary materials leave much to be desired. The code repository was incomplete - required dependencies and their versions were not specified; the dataset used was not supplied, and nor was any code supplied to create a dataset, making it impossible to replicate the results.

Relation to Prior Work: Yes.

Reproducibility: Yes

Additional Feedback: I thank the authors for their responses to queries.


Review 3

Summary and Contributions: The paper proposes joint policy search (JPS), a local policy improvement algorithm for teams of agents. The paper includes both a collection of small scale experiments and more significant experiments on contract bridge.

Strengths: Local policy improvement in imperfect information games is a hard problem carrying significant interest from the community. The performance of JPS on large scale games is impressive.

Weaknesses: The reviewer's most significant criticisms regard the paper's presentation. The first sentence states that DRL has demonstrated strong or even super-human performance in many games and gives examples. But many of the examples are neither deep nor reinforcement learning (e.g. Libratus, Pluribus). Deepstack, which is at least deep, is excluded. Is the paper suggesting that DeepStack did not demonstrate strong performance? The paper begins using notation in Section 3 before introducing it. The paper sometimes uses language without specifying its meaning, e.g. candidate infosets. The paper would be easier to read if it walked the reader through the pseudocode in Algorithm 1 and the decomposition in equations 8 & 9. A number of figures are not referenced in the main body of the paper. The graph in Figure 2 should use vector graphics. There is a typo in the first sentence of the proof of Lemma 2.

Correctness: The theory and empirical methodology both appear to be correct.

Clarity: Yes, although there are some issues.

Relation to Prior Work: Yes.

Reproducibility: Yes

Additional Feedback: Unlike for common-payoff games, there may be no pure optimal joint policy for Contract Bridge. It would be nice for the paper to comment on the practical implications (or lack there of) for JPS in this context. POST REBUTTAL: There are some issues with the submission: 1) Unfair comparison to WBridge5 2) Disinterest in acknowledging weaknesses of approach 3) Unclear relationship to subgame decomposition 4) Gap between theoretical results (cooperative games) and application (mixed cooperative/competitive game) Nevertheless, the empirical results are impressive and the submission should be given credit for its ambition.


Review 4

Summary and Contributions: The authors aim to show that in collaborative games (where all players share the same reward) the overall impact of policy changes on the team's game value can be decomposed into a summation over changes at altered information sets. The authors introduce the Joint Policy Search (JPS) algorithm, provide a theorem showing that JPS has a performance guarantee (it will find policy changes do not worsen the team's value), and empirically evaluate the algorithm relative to other approaches on a collection of toy games. Notably, they also demonstrate empirical results on contract bridge bidding, improving state-of-the-art performance against a championship-winning computer bridge agent, WBridge5.

Strengths: The paper is relevant to the NeruIPS community. The authors focus on imperfect-information extensive-form games, which have seen considerable research, past and current, in the NeurIPS community and elsewhere. The particular setting of fully collaborative games also makes this work strongly connected to research on Dec-POMDPs. The theoretical foundations of the work, namely identifying how to decompose the impact of a policy change, provides justification for their approach and characterization for the broader class of problems. The empirical results in contract bridge are particularly interesting.

Weaknesses: There are a couple weaknesses in the paper that I think the authors could improve. First, while this particular type of value decomposition and the JPS based on it are somewhat novel, the policy-change density that the authors claim to be novel bears considerable undiscussed similarity to the regret notions used by Zinkevich et al.'s CFR algorithm. Specifically, the definition of policy-change density in equation 6 resembles the regrets in equation 7 of "Regret Minimization in Games with Incomplete Information": CFR seems to just use a counterfactual weighting instead. The authors note later that they only consider pure strategies changes, which is also aligned with the per-action regret used in CFR. While I'm not overly concerned about the novelty claim, I do believe it is a missed opportunity to highlight such connections, particularly as their empirical results make a comparison with CFR algorithms. Similarly, while the authors note on line 41 that "changing policy on one decision point leads to reachability changes of downstream states, leading to non-local interplay between policy updates.", they do not connect these decomposition challenges to those of imperfect information games in general. In particular, work by Burch showed that for equilibrium construction, the problem is even worse than described, with behaviour at an information set being entangled across the game tree, not just due to downstream reachability. This discovery was behind algorithms like CFR-D (see Burch, Johanson, and Bowling in "Solving Imperfect Information Games Using Decomposition" (AAAI-14) and Burch's "Time and Space: Why Imperfect Information Games are Hard" (PhD Thesis)) and the continual resolving used by Moravcik et al. in "DeepStack: Expert-Level Artifical Intelligence in Heads-Up No-Limit Poker" (Science, 2017), both of which are unmentioned. Second, the paper's theoretical results seem to be limited to the case of fully collaborative games. As the paper's main evaluation domain of contract bridge is not fully collaborative, it is unclear to me if the claimed performance guarantee for JPS holds even there. While there are certainly fully collaborative domains that benefit from such analysis, I also have some concerns about their correctness (discussed later).

Correctness: The biggest concern that I have regarding correctness is in the proof of the paper's theoretical results. Specifically, in the proof of Lemma 2 (Appendix J.2), in the last equality for equation 15 the authors connect the sum back to \sum_{z \in Z} c(z). However, by equation 3, c^{\sigma, \sigma'}(h) = (\pi^{\sigma'}(h) - \pi^{\sigma}(h)) v^{\sigma}(h), which does not include the conditioning of having reached h_0. Why are these equivalent? If h_0 is the root of a subtree, then unless h_0 is the empty history, I don't see why these are the same. Perhaps there is an argument for why this works, but if there is then I believe the authors should provide it. I also feel the authors neglect to note some important context about their result against WBridge5. Due to space limits here, I expand on this and provide some other minor comments in the additional comments section.

Clarity: I believe the paper's clarity could be improved with some relatively small changes to how terms are used. On first reading, some terms were vague enough that they caused confusion about the correctness and scope of the work. The authors should be very clear with terms closer to the start of the paper. Primarily, it often becomes unclear what specific setting the authors are describing or what context their results hold in. The authors sometimes refer to "collaborative games", while at other times they refer to "fully collaborative settings" (as at the start of Section 3). Do they always mean fully collaborative (i.e., "where all agents share a common game value")? Because the authors also mention Bridge as an evaluation domain and use hidden cards in Poker as to illustrate information sets. While this is fine, I found it created some confusion when trying to follow their exposition and contextualize their results. For example, on line 51 the authors state "JPS is proven to never worsen the current policy". However, at this point in the paper it isn't clear what "collaborative games" specifically means, or even what performance measure this refers to (Performance against a fixed policy? Exploitability relative to a Nash equilibrium?). While this becomes more clear later in the paper, it would benefit readers to be aware of the subtext of these claims as they encounter them. The mixing of exposition based on "fully collaborative" games and Bridge made it unclear if the authors intended the joint policy search algorithm to have any guarantees in Bridge, since it is not a purely collaborative game. Some other minor comments regarding clarity are below. [Section 3, line 105] "later re-enter another active infoset I4 (Fig. 1(b))" > Perhaps it is worth noting that the example that you show in Fig. 1(b) can only occur in an imperfect-recall extensive-form game. [Section 3, line 130] "As one key difference between CIG and IIG..." > Did you mean v^{\sigma}(I) instead of V^{\sigma}(I)? [Table 1] "The super script ∗ means the method obtains best known solution." > I'm not sure what the asterisk is supposed to mean, as it is applied to multiple techniques with considerably different values, several of which are also below the "Best Known" value. [Table 2] > What are the +/- values? Standard error? Confidence intervals? Something else?

Relation to Prior Work: As discussed in the weaknesses section, I think highlighting connections to existing works is one of the areas the paper could improve on. In addition to previous comments, I would add that the discussion of related work in Section 2 reads more as a list of prior works and the domains that were addressed rather than a contrast with those works. While this phenomenon is not unique to this paper, I believe it would benefit readers to understand more about the limitations of those methods in this setting, particularly for the methods that are later used in the empirical results. Other comments regarding characterization of prior work are below. [Introduction, line 30] "assuming stationary environment and fixed policies of all other agents." > This isn't quite true. For example, to compute a Nash equilibrium for poker, agents are typically trained in self-play where all players are simultaneously learning. [Related work, line 69] "These algorithms are coordinate-ascent: iteratively find a best response to improve the current policy, given the opponent policies over the history." > CFR is not iteratively computing a full best response to the opponent. This is particularly clear to see when considering any of the Monte Carlo variants of CFR, which draw samples according to a current policy and update the policy based on regret minimization. [Related work, line 86] "With Deep CFR [6], DeepRole achieves" > It seems like a citation to the DeepStack work is also warranted here, as that is what the DeepRole paper itself said it was building upon, and the DeepStack work introduced several concepts that the Deep CFR paper later built upon.

Reproducibility: Yes

Additional Feedback: Regarding the correctness of the empirical methodology, the evaluation against WBridge5 lacks important context that perhaps the authors do not realize. Footnote 1 states "Unlike hint games like Hanabi, public actions in Bridge (e.g. bid) do not have pre-defined meaning and does not decrease the uncertainty when game progresses". While the authors are correct that the bids do not have an explicit grounded meaning within the game, my limited knowledge of Bridge conventions is that such bids actually do have meaning that is specifically pre-defined and must be disclosed. For example, from Wikipedia (https://en.wikipedia.org/wiki/Bridge_convention): "Conventions to be played must be agreed by partners before play begins and must be disclosed to their opponents, either in advance by the use of convention cards or by alerts, announcements, and answers to questions about one's partner's bids once bidding has begun. Generally, this disclosure also must include the negative implications of choosing the bid over another alternative. Failure to reveal fully the existence and meaning of a convention generally constitutes an illegal communication of information between partners." Although I'm unfamiliar with details of WBridge5's specific conventions, I expect that these conventions are well-documented and allowed by the wider bridge community. While it is impressive to beat WBridge5, I have to wonder if this is a misleading comparison, with WBridge5 forced to comply with conventions while JPS is allowed to deviate to convention systems that WBridge5 and the wider bridge community disallow. At the very least, I believe the authors should be noting this relevant context. The remaining comments that I have about correctness are relatively minor. [Related work, line 65] "complete information game" > I think you mean "perfect information". Similarly, for IIG, I think you mean "imperfect information games". Note that in game theory, complete and perfect information are two (unfortunately named) distinct game properties, where complete information games can still have imperfect information. [Section 3, line 98] "Obviously, we could enumerate all policies changes" > Why is this true? The policy space is continuous, and not necessarily enumerable. Are you assuming the agents act deterministically? [Section 3, line 121] > Is this "reachability" intended to only capture the sequence probability for the agent acting according to \sigma? What about chance? Or the other players? Do the other players also act according to \sigma? [Section 3.2, line 137] "Intuitively, c(h) means if we switch from \sigma to \sigma', what would be the difference in terms of expected reward, if the new policy \sigma' remains the same for all h’s descendants." > Restricting \sigma' for only the descendants seems insufficient. This means that \sigma' could be different prior to h, which certainly impacts the expected reward of \sigma' overall. Do you specifically mean the difference in expected reward from the subtree under h? [Section 3.2, lemma/theorem proofs] > There is no mention that there are proofs in your appendix. This should be added. [Section 3.3, line 166] "Dominated by pure strategies, mixed strategies are not considered." > While this might be true for purely cooperative settings, I see no reason mixed strategies would be dominated in general imperfect information games with only some agents cooperating. Maybe "Dominated by pure strategies, mixed strategies are not considered for our purely cooperative settings." [Section 3.3, Theorem 2] > The argument made so seems to rely on this being a fully collaborative game. Considering you apply these techniques to Contract Bridge, you should remind readers that these guarantees do not hold in that setting. Finally, some miscellaneous minor comments. [References] > I would update citations that reference arxiv publications when they were subsequently published after peer review (e.g., BAD was at ICML 2019). [Introduction, line 54] "Advantageous Actor-Critic" > "Advantage Actor-Critic" [Related Work, line 92] "Recent works use DRL" > Personal preference, use Deep RL instead of DRL if you have to shorten it. ====================== === Post author rebuttal === ====================== Thank you for your comments. I think it is important to be specific and contextualize any claim made relative to WBridge5 with the caveats you point out in your rebuttal. In particular, while you note the unfair comparison in lines 251-253, this exposition did not make clear the distinction that JPS can arrive at illegal conventions while WBridge5 uses valid ones. While the result remains interesting despite this, for readers who are unfamiliar with the details of bridge, this nuance will be wholly non-obvious and highly relevant context. Regarding your response on generality. I would discourage including claims about extending this to general-sum games in the final version of the paper as it is not clear to me that you can currently substantiate them. In particular while pure strategies are sufficient in fully cooperative games, as there is no need for information hiding, pure strategies do not dominate mixed strategies in general-sum games. While this may not have a material impact on performance in bridge, games like poker are general-sum games and have well documented examples of mixed strategies being required for equilibrium strategies. This makes Algorithm 1 ill-defined as you can't enumerate candidate policy changes. Furthermore, the results from CFR-D suggest that decomposing a game at the resolution of information sets has demonstrable failure modes. With Algorithm 1 being ill-defined in this setting, do you retain the does-no-harm guarantee from Theorem 2? If you wanted to make claims on general-sum games, I feel that you would need to do considerably more to explain why such a result is possible in light of the CFR-D findings.

[Author Response · NeurIPS 2020]

We thank reviewers (R1,R2,R3,R5) for their insightful comments. All reviewers agree that the main quantity proposed in the paper, *policy-change density*, is novel and the proposed decomposition of policy change over active information set, can be potentially impactful for research in incomplete information games (IIG), in which unlike perfect information game, policies at different information sets can influence each others in involved ways. Most reviewers agree that experiments on simple games and Bridge bidding are interesting.

We thank R5 for pointing out that the "decomposition challenges" in IIG are critical for equilibrium construction where the problem is "even worse than described, with behaviour at an information set being entangled across the game tree, not just due to downstream reachability." Therefore, our paper could have stronger implications than we expect. We will make connections to CFR-D, continual resolving, etc, which are for subgame solving in zero-sum 2-player games. We disagree with R2 that the tabular form of JPS indeed has theoretical guarantees, as appreciated by other reviewers.

**Generality**. Our framework can be extended to general-sum games, by replacing the scalar value $v^\sigma$ with vector values $\mathbf{v}^\sigma \in \mathbb{R}^C$, where $C$ is the number of players. The $i$-th component of $\mathbf{v}^\sigma$ is the utility of player $i$. Other terms (e.g., $c^{\sigma,\sigma'}$ and $\rho^{\sigma,\sigma'}$) are based on $v^\sigma$ and can be similarly vectorized. In this paper we dig deep in pure collaborative settings and apply JSP for competitive Bridge bidding to improve the collaboration within the team, under the self-play framework.

**Comparison with WBridge5** R2,R5. We are aware of the potential unfairness of comparing with WBridge5 only at Bridge bidding phase (line 251-253), including (**1**) WBridge5 conforms to human convention but JPS can be creative, (**2**) WBridge5 optimizes for the results of real Bridge playing rather than double-dummy scores (DDS) that assumes full information during playing. While we will *tune down the claim* as suggested by R2,R5, fully addressing these problems requires substantial future work, as R2 points out. For now, to verify our bot, we choose to evaluate against WBridge5, which is an independent baseline tested extensively with both AI and human players. Full game AI is a future work.

**Connection to CFR**. R5 makes a great point that similarity exists between our policy-change density (Eqn. 6 in our paper) and regret notation (Eqn. 7 in[1]). The key difference here is that we use the total reachability $\pi^{\sigma'}$ evaluated on the *new* policy $\sigma'$, while CFR uses the except-player-$i$ reachability $\pi^\sigma_{-i}$ evaluated on the *old* policy $\sigma$.

This change leads to very different (and novel) theoretical insights. It leads to policy-change decomposition in Thm. 1, and enables the proposed search algorithm (Alg. 1) with guarantees. Furthermore, our decomposition formula *exactly* captures the value difference before and after policy changes, while in CFR, summation of the regret notion is *an upper bound* of the Nash exploitability. Our advantage comes with a price: the regret in CFR only depends on the old policy $\sigma$ and can be computed independently at each infoset, while computing our policy-change density requires a re-computation of the altered reachability due to new policy $\sigma'$ on the upstream infosets. From the derivation, we could also see that CFR requires the condition of *perfect recall* to ensure that no double counting exists so that the upper bound can hold (Eqn. 15 in[1]), while our formula does not require that. We will add comparisons in the next version.

**Concern in correctness of Lemma 2**. R5 expresses "biggest concerns" that in Eqn. 15, the quantity $c(z)$ is defined differently than its original definition (Eqn. 3, line 136). Due to an editing error, the last equality "$= \sum_{z \in Z} c(z)$" in Eqn. 15 is unnecessary and shouldn't appear. As suggested by R3, there is a typo in Eqn. 14: it should be "$v^{\sigma'}(h_0) = \sum_{z \in Z} \pi^{\sigma'}(z|h_0)v(z)$". The remaining part of Lemma 2 is consistent. We apologize for editing errors. Experiments show that the game value difference $\bar{v}^{\sigma'} - \bar{v}^\sigma$ from Thm. 1 always coincides with naive computation, with much faster speed. [R1] E.g., for each iteration in SimpleBidding (Def. 2), for $N = 8$ JPS takes $\sim 1$s while brute-force takes $\sim 4$s; for $N = 16$ and $d = 3$ JPS takes $\sim 20$s while brute-force takes $\sim 260$s.

**Bidding conventions**. As said by R2 and R5, by Law 40 of Contract Bridge[2], before the game starts both team needs to fully disclose the meaning of their bids and answer questions accordingly. In Computer Bridge Tournament, conventions needs to be disclosed 1 month beforehand for other bots to adapt[3]. For WBridge5, its convention is fixed[4] and it seems that a manual change of its internal logic is needed, which is not possible without the code.

**JPS on other policies [R1]**. Except for Comm (Def. 1) that JPS always gets 1.0, uniform random+JPS converges to local minima that CFR is immune to, and underperforms CFR1k+JPS. Compared to CFR1k+JPS, BAD+JPS is worse (10.47 vs 10.56 for $N = 16$) in Simple Bidding but *better* (1.12/1.71/2.77 vs 1.07/1.71/2.74 for $N = 3/4/5$) in 2-Suited-Bridge. We leave these interesting interplays between methods for future study. Except for brute-force and JPS, we are not aware of other methods with the same non-worsening guarantee for policy update in cooperative IIGs.

**Other issues.** [R1] *"Percentage of Search"*: in non-tabular version of JPS, estimation of $v^\sigma$ runs in parallel with Alg. 1. The percentage determines the ratio of threads running estimation to those running Alg. 1. Direct comparison with [18] is not possible due to unpublished code. [R5] In performance like "$\mu \pm \sigma_\mu$", $\sigma_\mu$ is the standard deviation of the estimated mean $\mu$. $\sigma_\mu \sim 1/\sqrt{N}$ for $N$ samples. In Table 1, all $\sigma_\mu$ are small ($\sim 10^{-2}$) and omitted. Superscript $*$ = one of the trials gets the best solution. *"Reachability"*: $\pi^\sigma$ includes chance and all agents playing under $\sigma$. We will fix all typos, add more citations and discussions and release the code with detailed instructions in the next revision.

## Footnotes

[1] https://poker.cs.ualberta.ca/publications/NIPS07-cfr.pdf    [2] https://www.acbl.org/conduct-and-ethics/active-ethics/#Disclosure

[3] https://www.allevybridge.com/allevy/computerbridge/icgaj.html    [4] https://github.com/jdh8/wbridge5.book


[Meta-Review · NeurIPS 2020]

This paper presents the concept of policy density change for collaborative imperfect information games. All the reviewers agree that the idea is novel, appreciating the results in small games and in a much larger game of bridge (in particular, a comparison vs. WBridge5). There are several problems identified that the reviewers agree to be characterized as minor enough to be address in the final copy. As noted, there are problems with the comparison to WBridge5 and the authors have agreed to change their claim as a result. Clarifications on the connections to CFR and subgame decomposition should be made. Finally, several reviewers brought up that authors seem dismissive or reluctant to admit the weaknesses of their approach. This could affect the long-term impact of the paper. In particular, in the discussion the reviewers seem skeptical that the approach could be as easily extended to the general-sum case as implied in the author response, and stated that such speculation could actually be harmful to the paper's narrative. I strongly encourage the authors to consider this feedback when improving the paper